# Juglone Inactivates *Pseudomonas aeruginosa* through Cell Membrane Damage, Biofilm Blockage, and Inhibition of Gene Expression

**DOI:** 10.3390/molecules26195854

**Published:** 2021-09-27

**Authors:** Qiqi Han, Xinpeng Yan, Runguang Zhang, Guoliang Wang, Youlin Zhang

**Affiliations:** College of Food Engineering and Nutrition Science, Shaanxi Normal University, Xi’an 710119, China; hqq@snnu.edu.cn (Q.H.); echoyan@snnu.edu.cn (X.Y.); sunshine@snnu.edu.cn (R.Z.)

**Keywords:** *P. aeruginosa*, juglone (5-hydroxy-1; 4-naphthoquinone), cell membrane, ROS, RT-qPCR

## Abstract

Due to the strong drug resistance of *Pseudomonas aeruginosa* (*P. aeruginosa*), the inhibition effects of conventional disinfectants and antibiotics are not obvious. Juglone extracted from discarded walnut husk, as a kind of plant-derived antimicrobial agent, has the advantages of naturalness, high efficiency, and low residue, with a potential role in the inhibition of *P. aeruginosa*. This study elucidated the inhibitory effect of juglone on the growth of plankton and the formation of *P. aeruginosa* biofilm. The results showed that juglone (35 μg/mL) had an irreversible inhibitory effect on *P. aeruginosa* colony formation (about 10^7^ CFU/mL). The integrity and permeability of the cell membrane were effectively destroyed, accompanied by disorder of the membrane permeability, mass leakage of the cytoplasm, and ATP consumption. Further studies manifested that juglone could induce the abnormal accumulation of ROS in cells and block the formation of the cell membrane. In addition, RT-qPCR showed that juglone could effectively block the expression of five virulence genes and two genes involved in the production of extracellular polymers, thereby reducing the toxicity and infection of *P. aeruginosa* and preventing the production of extracellular polymers. This study can provide support for the innovation of antibacterial technology toward *P. aeruginosa* in food.

## 1. Introduction

Food-borne bacterial pathogens pose a serious threat to human health, causing millions of deaths each year, and the proportion in developing countries is even higher [1]. As one of the common foodborne pathogens, *Pseudomonas aeruginosa (P. aeruginosa)* has strong drug resistance, infection, and pathogenicity, and it can easily harm human health by contaminating water and food. *P. aeruginosa*, which causes many infections in the world each year, especially in patients with burns and lung cysts, is currently being monitored in many cities in China [2]. In order to combat food-borne pathogens, humans have synthesized a large number of fungicides, antibiotics, and chemical preservatives. Unfortunately, the continuous and widespread use of antibiotics and chemicals has led to the frequent emergence of drug-resistant bacteria, eventually causing serious human health problems [3,4]. Therefore, in order to effectively avoid the development of drug-resistant bacteria, screening natural green antimicrobial agents from rich natural products has become the focus of most in the field of antibacterial technology innovation toward food-borne bacterial pathogens.

*P**. aeruginosa*, as a Gram-negative aerobacterium, is a food pathogen commonly found in water and beverage products (including pasteurized milk) [5]. *P. aeruginosa* can quickly cross the defense barrier of the host cell and produce severe pathological effects on human body, such as an increase in the incidence rate and mortality of cystic fibrosis patients, and severe infection symptoms including septicemia, pneumonia, otitis media, purulent wounds, urinary tract infections, and bacteremia [6]. Its strong resistance to antibiotics and the strong adaptability of its cell membrane to external stress leads to a high infection rate of *P. aeruginosa* in the human body. Unfortunately, traditional disinfectants and antibiotics have no obvious inhibitory effect on *P. aeruginosa*. Therefore, the development of natural antimicrobial agents against *P. aeruginosa* has become the focus of researchers.

As they are harmless to humans, the application of natural preservatives of plant origin has become popular in the food industry [7]. Juglone, extracted from walnut tree, has been widely used in the pharmaceutical industry, cosmetics industry, and food industry as an additive to enhance the color and flavor of food [8,9]. Furthermore, juglone has great potential in inhibiting bacterial growth, along with anticancer, antivirus, and anti-inflammatory properties [10,11,12]. In particular, juglone can inactivate *Staphylococcus aureus* and *Escherichia coli* via the destruction of bacterial biofilm, inhibition of enzyme expression, destruction of nucleic acids, and other pathways [13]. Furthermore, our previous study also found that juglone has a significant inhibitory effect on food-borne pathogens.

However, there are few studies on the inhibitory effect of juglone on *P. aeruginosa*. Accordingly, this paper evaluates the antibacterial mechanism of juglone against *P. aeruginosa* from the aspects of cell-membrane destruction, induction of an abnormal accumulation of ROS, and inhibition of gene expression.

## 2. Materials and Methods

### 2.1. Materials and Culture

*P. aeruginosa* ATCC 10145 strains were obtained from the Food Safety and Hygiene Laboratory (Shaanxi Normal University, Xi’an, China). All other reagents used in the experiment were analytically pure. Juglone was extracted from walnut husk and purified to 70.5% purity (Walnut husk obtained with Xi ′an Zhuque market). Before each experiment, the *P. aeruginosa* strain was cultured at 37 °C for 16 h [5].

### 2.2. Extraction, Purification of Juglone, and Content Analysis

Walnut husk from China′s Shaanxi “xiangling” variety of walnut, specially purchased from Xi’an rosefinch market. The collected walnut husk was frozen for 2 h, freeze-dried for 72 h, crushed to powder, and stored at −20 °C. Next, 5 g of powder was soaked in chloroform (1:15, *w*/*v*), placed in a 30 °C water bath, and ultrasonicated at 100 W for 40 min in a fume hood. Then, the crude product was concentrated by rotating, dissolved in methanol, and then freeze-dried for 48 h. The crude juglone powder was obtained.

The crude extract of juglone was diluted with methanol to 0.311 mg/mL, before adding 30 g of HPD-100 (Haoju Company, Tianjin, China) macroporous resin. The sample was shaken at 25 °C for 2 h at 190 rpm. The saturated macroporous resin was washed with distilled water three times, loaded into a chromatographic column (60 cm × 1.5 cm), and rinsed with chloroform at a desorption rate of 8 drops/min.

The purified juglone was dissolved in methanol and determined by HPLC. The conditions of the HPLC method were as follows: Agilent C18 column, mobile phase methanol–water (55:45 (*v*/*v*)), injection volume 10 μL, volume flow rate 1 mL/min, column temperature 30 °C, detection wavelength 250 nm [14].

### 2.3. Antibacterial Activity against P. aeruginosa

#### 2.3.1. Determination of Bacteriostatic Zone Diameter (DIZ)

Further improvement was made by referring to the method of Liu et al. [5]. Prior to the experiment, the activated *P. aeruginosa* strain was cultured in MH broth at 37 °C for 16 h in a shaking flask and diluted with MB broth until the OD value was approximately 0.5 (approximately 10^7^ CFU/mL). The suspension of 200 μL of bacteria (approximately 10^7^ CFU/mL) was evenly spread on the Mueller–Hinton (MH) agar, and an Oxford cup was placed in the center of the plate. Then, 150 μL of 70 μg/mL juglone (the concentration of dissolved DMSO was 0.5% (*w*/*v*)) was added to the Oxford cup and incubated at 37 °C for 24 h.

#### 2.3.2. Determination of Minimum Inhibitory Concentration (MIC) and Minimum Bactericidal Concentration (MBC)

The juglone was dissolved in MH broth to final concentrations ranging from 5 μg/mL to 55 μg/mL. Then, 50 μL of bacterial suspension (approximately 10^7^ CFU/mL) was inoculated into each tube and cultured at 37 °C for 24 h. The MIC was considered to be the minimum concentration of juglone, and the medium was clear and transparent. The bacterial suspension prepared in the test tube was coated on the MH agar and cultured at 37 °C for 24 h. If no bacterial growth was observed on the MH agar, the MBC value was defined. Gallic acid was used as a positive control, and the determination method was similar to the above.

### 2.4. Measurement of Growth Curve

To the experimental method of Kang et al. [15] was referenced. Juglone was dissolved in LB medium to final concentrations of 0, 1/8MIC, 1/2MIC, MIC, and MBC. Then, the bacterial suspension (approximately 10^7^ CFU/mL) was inoculated into the test tubes of each experiment and incubated on a shaker at 37 °C. The test tube was taken out every 2 h for 24 h, and the OD (wavelength of 600 nm) value of bacteria was measured using a spectrophotometer (1530, Thermo Fisher Scientific Oy, FSP GROUP INC, Wuhan, China).

### 2.5. Antimicrobial Mechanism of Juglone against P. aeruginosa

#### 2.5.1. Confocal Laser Scanning Microscopy (CLSM)

The bacterial viability test was similar to the previous study by Kang et al. [16]. The bacterial suspension (approximately 10^7^ CFU/mL) was centrifuged, the supernatant was discarded, and the precipitated cells were resuspended with sterile PBS. Then, *P. aeruginosa* cells were treated with juglone at the final concentration of 0 (control), 1/2MIC, MIC and MBC, before incubating at 37 °C for 2 h. Next, 4 mL of the mixture was centrifuged and washed for precipitation to obtain cells. The precipitate was resuspended with 500 μL of PBS, and 20 μL of propidium iodide (PI) and SYTO9 dye were added. The precipitate was stained in the dark for 15 min. Finally, the sample was analyzed using an FV1200 confocal laser scanning microscope (Olympus Corporation, Tokyo, Japan) with an objective of ×40.

#### 2.5.2. Field-Emission Scanning Electron Microscopy (FE-SEM)

Further modifications were made of the method according to Ning et al. [17]. First, 4 mL of *P. aeruginosa* suspensions (approximately 10^7^ CFU/mL) were treated with juglone (at final concentrations of 0 (control), 1/2MIC, MIC, and MBC), before incubating at 37 °C for 4 h. The cells were obtained by centrifuging and washing with 6 mL of bacterial suspension. The precipitate was resuspended with 1 mL of 2.5% glutaraldehyde and fixed at 4 °C for 4 h. The precipitate was eluted with a concentration gradient (30%, 40%, 50%, 60%, 70%, 80%, 90%, 100%) of ethanol for 10 min. The sample was then mixed with *tert*-butanol and freeze-dried. Finally, the dehydrated samples were sprayed with gold and observed and analyzed under an FE-SEM (SU8220, Hitachi, Tokyo, Japan).

#### 2.5.3. Determination of Cell Membrane Fluidity

The method for determining the fluidity of the cell membrane of *P. aeruginosa* was similar to previous studies [18], but with some modifications. First, 4 mL of *P. aeruginosa* suspensions (approximately 10^7^ CFU/mL) were treated with juglone (at final concentrations of 0 (control), 1/2MIC, MIC, and MBC), before incubating at 37 °C for 4 h. Then, 1 mL of sample was centrifuged at 6000 rpm for 10 min to obtain cells, and the obtained cells were fixed in 0.37% formaldehyde solution for 30 min. The bacteria were frozen in liquid nitrogen for 5 min, thawed, resuspended in 0.6 mM DPH solution, and incubated at 28 °C for 45 min. Finally, the fluorescence intensity of DPH was measured using a fluorescence spectrophotometer (the excitation wavelength was 350 nm and the emission wavelength was 425 nm).

#### 2.5.4. Whole-Cell Protein Analysis and Determination of Protein Leakage

First, 10 mL of *P. aeruginosa* (approximately 10^7^ CFU/mL) cells treated with juglone (at a final concentration of 0 (control), 1/2MIC, MIC, and MBC) for 12 h were collected and resuspended in 1 mL of PBS, before treating with ultrasound for 5–10 min in an ice bath. The cleaned sediment was treated with a bacterial protein extraction kit (Solarbio, Beijing, China) to obtain bacterial intracellular proteins. The extracted protein was mixed with the protein dye in equal amounts, and then the mixture was boiled for 5 min, before being centrifuged at 10,000 rpm for 5 min. According to the protocol of the SDS-PAGE gel preparation kit (SolarBio, Beijing, China), 5% stacking and 12% separation gels were used for electrophoresis, before dyeing with Coomassie Bright Blue for 2 h. Finally, the gel was rinsed with eluent for another 10 h.

At 0, 2, 4, 6, 8, 10, and 12 h, 1 mL of bacterial suspension was centrifuged to obtain supernatant. The protein content in the supernatant was determined using the protein detection kit (Beyotime Biotechnology, Shanghai, China) to determine the damage to the *P. aeruginosa* membrane.

#### 2.5.5. Intracellular and Extracellular Nucleic Acid Leakage Analysis and DNA Gel Electrophoresis

According to the method of Liu et al. [5], juglone (at final concentrations of 0 (control), 1/2MIC, MIC, and MBC) was added to 5 mL of *P. aeruginosa* (approximately 10^7^ CFU/mL) and incubated for 30 min. Subsequently, a 1 mL bacterial suspension was centrifuged at 8000 rpm for 5 min, and the absorbance of the supernatant was measured at 260 nm wavelength. At the same time, the intracellular nucleic acids were obtained by treating the precipitate with a DNA extraction kit (Sangon, Shanghai, China). The resulting DNA was diluted to a specific concentration with Tris-HCl (0.1 mol/L, pH 8.0). Then, 5 μL of extracted DNA was mixed with 1 μL of buffer solution. The mixture was subjected to agarose gel electrophoresis for 15 min (220 V).

#### 2.5.6. Determination of Intracellular ATP Concentration and ATPase Activity

*P. aeruginosa* (approximately 10^7^ CFU/mL) treated with juglone (at final concentrations of 0 (control), 1/2MIC, MIC, and MBC) for 30 min was collected and washed with PBS. PBS was used to resuspend the bacteria, with the concentration of the bacteria reaching 10^6^ CFU/mL. Then, according to the steps of the ATP extraction kit (Jiancheng, Nanjing, China), the intracellular ATP was extracted, and its activity was detected. The activity of Na^+^K^+^-ATPase in cells (Jiancheng, Nanjing, China) was detected according to the procedure of the ATPase detection kit.

### 2.6. Reactive Oxygen Species (ROS) Analysis

ROS content was detected using the DCFH-DA fluorescence kit (Jiancheng, Nanjing, China). DCFH-DA (2′,7′-dichlorodihydrofluorescin diacetate) is a general indicator of oxidative stress with membrane permeability and no color of its own. After entering cells, DCFH-DA is hydrolyzed by cellular lipase to produce DCFH (2′,7′-dichlorodihydrofluorescin) and then rapidly oxidized to produce the strong fluorescence product DCF (2′,7′-dichlorofluorescin), which was finally detected by fluorescence spectroscopy. Referring to Ning et al.’s [17] method, the specific operations were as follows: juglone (0 (control), 1/2MIC, MIC, and MBC) was mixed with 4 mL of *P. aeruginosa* (approximately 10^7^ CFU/mL) and cultured at 37 °C for 3 h. Then, a 3 mL bacterial suspension was centrifuged to obtain precipitate cells. After the suspension was resuspended with PBS, 49:1 DCFH-DA was added, before incubating at 37 °C in darkness for 30 min. Finally, the reactive oxygen species level in the mixture was detected by flow cytometry (BeamCyte, Changzhou, China).

### 2.7. Inhibition of Biofilm Formation by Juglone

#### 2.7.1. Biofilm Formation

The indices of inhibiting biofilm formation were determined using the method of Liu et al. [5]. Different concentrations of juglone (at final concentrations of 0 (control), 1/8MIC, 1/4MIC, 1/2MIC, MIC, and MBC) were added to the bacterial suspension. The mixed samples were then added to the 96-well polystyrene plate and incubated at 37 °C for 24 h. After the biofilm was formed, each well was cleaned with PBS, and then fixed with methanol for 15 min. Subsequently, 1% crystal violet was added to each well and left for 15 min. Finally, it was dissolved in 95% ethanol and evaluated in a microplate analyzer (OD at 570 nm). The measured values were imported into the following formula to calculate the percentage inhibition of biofilm formation:(1)$$=\frac{{\mathrm{OD}}_{570}\left(\mathrm{control}\right)-{\mathrm{OD}}_{570}\left(\mathrm{sample}\right)}{{\mathrm{OD}}_{570}\left(\mathrm{control}\right)}\times 100\%.$$

#### 2.7.2. Determination of Living Cells

A *P. aeruginosa* suspension (approximately 10^7^ CFU/mL) and medium containing different concentrations of juglone (at finial concentrations of 0 (control), 1/8MIC, 1/4MIC, 1/2MIC, and MIC) were added to a six-well polyethylene plate. After incubation for 48 h, the planktonic cells were cleaned and resuspended with PBS, before being mechanically dispersed to destroy the biofilm. The plate counting method was used to quantitatively analyze the biofilm [16].

#### 2.7.3. ESEM Morphological Analysis of Biofilm

Samples were processed according to the research methods of Kang et al. [15]. A *P. aeruginosa* suspension (approximately 10^7^ CFU/mL) and juglone (0, 1/4MIC, 1/2MIC, and MIC) were added to LB broth and incubated at 37 °C for 24 h to form biofilms on glass coverslips (4 mm × 4 mm). Subsequent treatment was the same as that in Section 2.5.2. Finally, the samples were sprayed with gold, and the cell membrane morphology was observed under ESEM.

#### 2.7.4. Determination of Bacterial Metabolic Activity in Biofilms

A *P. aeruginosa* suspension (approximately 10^7^ CFU/mL) and medium containing different concentrations of juglone (0, 1/8MIC, 1/4MIC 1/2MIC, and MIC) were added to a 96-well polyethylene plate. The metabolic activity of the biofilm was determined using an MTT (3-(4, 5-dimethylthiazol-2-yl)-2, 5-diphenyltetrazolium bromide) assay kit. Finally, the spectrophotometric value was determined at 570 nm.

#### 2.7.5. Swimming and Swarming Motilities Assays

The swimming and swarming motilities of *P. aeruginosa* were studied by referring to Liu et al. [5]. The swimming motility of *P. aeruginosa* was evaluated by adding 0.3% agar to 20 mL of LB broth as a semisolid medium. Different concentrations of juglone (0, MIC, and MBC) were added before the medium was cured. After the medium was solidified, 5 μL of *P. aeruginosa* (approximately 10^7^ CFU/mL) was added and cultured at 37 °C for 7 h. The swarming test was similar to the swimming test except that the swarming medium was supplemented with 0.5% agar and 16% glucose. The culture was placed at 37 °C for 20 h, and the diameter was measured.

### 2.8. RNA Extraction and Real-Time Quantitative Polymerase Chain Reaction (RT-qPCR) Validation

*P. aeruginosa* treated with juglone (0, MIC, and MBC) for 12 h was collected and washed with PBS. Total RNA was isolated from the cells using the Trizol kit (Sangon, Shanghai, China) and reverse-transcribed into cDNA using the Hifiscript cDNA synthesis kit (Kangweishiji, Beijing, China). The quality and quantity of the cDNA obtained were determined by measuring the absorbance at 260 nm and 280 nm. Specific primers were used to detect the presence of five extranotoxin genes (Exou, Exos, Phzm, Toxa, Lasb), two key genes involved in the production of extracellular polymers (epsA, epsF), and the internal control gene (Ecfx) in the reverse-transcribed cDNA by multiple polymerase chain reaction (Table 1). The real-time PCR was performed using the UltraSYBR mixture (Kangweishiji, Beijing, China). The circulating conditions of the five exotoxin genes were as follows: reaction at 95 °C for 3 min, followed by 45 cycles of initial denaturation, with denaturation at 93 °C for 30 s, annealing at 53 °C for 30 s, extension at 72 °C for 50 s, and a final extension at 72 °C for 5 min. EpsA and epsF genes were produced under cyclic conditions as follows: reaction at 93 °C for 5 min, followed by 35 cycles of initial denaturation, with denaturation at 95 °C for 30 s, annealing at 53 °C for 30 s, extension at 72 °C for 50 s, and a final extension at 72 °C for 10 min. The relative gene expression was calculated using the 2^−ΔΔCt^ method [19,20]. The names and primer sequences of the five toxin genes, epsA, epsF, and reference genes are shown in Table 1.

### 2.9. Statistical Analysis

All experiments were set up in three parallel groups, and the data were expressed as the mean ± standard deviation. IBM SPSS (version 22.0; SPSS, Inc., Chicago, IL, USA) statistical software was used for statistical analysis. The fluorescence intensity of the DCFH-DA probe for *P**. aeruginosa* treated with different concentrations was detected by flow cytometry, and the production of intracellular reactive oxygen species was evaluated by the change in fluorescence intensity. The differences were calculated using one-way analysis of variance and were considered to be statistically significant at *p* < 0.05.

## 3. Results

### 3.1. Juglone Extraction Rate, Purity, and HPLC Content Analysis

According to the experimental calculation, the extraction rate of the crude juglone was 0.96%, and the purity was 30.12%. The adsorption rate of HPD-100 type macroporous resin was 86.54%, the elution efficiency was 97.44%, and the purity of juglone was 70.59%. The HPLC spectrum analysis results of juglone standard and juglone before and after purification are shown in Figure 1A–C. Each exhibited peaks at about 18 min, while the purity of the purified juglone was obviously higher than that of extracted juglone.

### 3.2. Antimicrobial Activity of Juglone and Its Effect on the Growth of P. aeruginosa

Figure 2A shows the chemical formula for juglone. The DIZ, MIC, and MBC of juglone against *P. aeruginosa* are shown in Table 2. The growth curve of *P. aeruginosa* is shown in Figure 2. The DIZ value of *P. aeruginosa* treated with juglone was 22.89 ± 1.00 mm, and the MIC and MBC were 35 μg/mL and 45 μg/mL, respectively. When the concentration of walnut ketone reached 1/8MIC, it began to have an inhibitory effect on *P. aeruginosa*, but the inhibitory effect was not obvious. Similar to the control group, the maximum growth concentration was reached at 12 h, and the difference in colony concentration was only about 0.7. With the gradual increase in juglone concentration, the MIC and MBC treatment groups had the most significant antibacterial effect. Compared with the control group, there was no obvious colony growth in the first 2 h, and this difference was extremely significant (*p <* 0.01) (Figure 2B). Compared with the killing effect of gallic acid on *P. aeruginosa*, juglone had an obvious advantage.

### 3.3. Effect of Juglone on Cell Viability

Figure 3 shows changes in the viability of *P. aeruginosa* culture caused by juglone. The untreated group was bright blue with no bright red spots, indicating that the cells in the untreated group had normal morphological and physiological functions. In the treatment group, as the concentration of juglone increased from 1/2MIC to MIC, the blue signal gradually decreased and the red signal gradually increased. When the concentration of juglone increased to MBC, all the blue signals turned red.

### 3.4. Effect of Juglone on the Morphology of P. aeruginosa

As shown in Figure 4, cells in the untreated group had smooth rod-like structures, full cells, and normal structures. In contrast, some cells of *P. aeruginosa* treated with 1/2MIC juglone were damaged, and their contents had leaked. When the concentration was increased to MIC, the cell membrane of most cells exhibited atrophy and damage, whereby the surface had collapsed and a large number of contents had leaked. Furthermore, as the concentration of juglone in the treatment group was increased to MBC, the cells no longer had a normal cell morphology and function.

### 3.5. Destruction of Cell Membrane of P. aeruginosa by Juglone

Figure 5 shows the effect of juglone on the *P. aeruginosa* cell membrane. DPH (C_9_H_9_NO) is a hydrophobic fluorescent reagent. When the phospholipid bilayer is damaged, DPH falls off from the phospholipid bilayer, resulting in a decrease in the fluorescence intensity of DPH [18]. Therefore, the fluorescence intensity of DPH can be used to measure the damage degree of the phospholipid bilayer of a cell membrane. Figure 5A shows that, after the treatment with 1/2MIC, MIC, and MBC concentrations of juglone, the fluorescence intensity of DPH showed a significant and gradual decreasing trend, indicating that the fluorescence intensity of DPH was negatively correlated with the amount of juglone added. There were significant differences between the treatment group and control group (*p* < 0.01). These results indicated that the addition of juglone prevented the insertion of DPH into the phospholipid bilayer of the membrane, thus inhibiting the fluidity of the membrane and destroying the integrity of the membrane structure.

Proteins and nucleic acids play an important role in maintaining normal bacterial cell growth, metabolism, cytoskeleton formation, and intracellular homeostasis. Once the cell membrane is damaged, the intracellular cytoplasm leaks. As shown in Figure 5B, there was no significant protein leakage in the control group throughout the incubation cycle. The MIC and MBC groups had sustained and significant protein leakage throughout the incubation cycle, reaching a peak value at around 8 h, with the leakage amounts reaching 140.5 μg/mL and 165.7 μg/mL, respectively. Compared with the control group, this difference was extremely significant (*p* < 0.01). The gel electrophoresis patterns of bacterial proteins were also consistent with the above results. Compared with the control group, the color of the bands in the administration group was significantly lighter, and the protein bands with molecular weight higher than 75 kDa disappeared in the MBC group (Figure 5C). As shown in Figure 5E, the intracellular DNA concentrations of *P. aeruginosa* were 259.9, 81.45, and 56.25 μg/mL after the addition of juglone at concentrations from ranging from 1/2MIC to MBC, respectively. The OD_260nm_ value increased from 0.012 to 0.033 μg/mL after the addition of juglone (Figure 5D). The results showed that the protein and nucleic acid leakage of *P. aeruginosa* increased with the increase in time and concentration, which intensified cell osmosis and disrupted the normal growth and metabolism of *P. aeruginosa*, eventually leading to cell death.

We further evaluated the intracellular ATP leakage of *P. aeruginosa* after exposure to juglone. Figure 5G shows the intracellular ATP concentration changes induced by juglone. The ATP content in *P. aeruginosa* treated with juglone was significantly lower than that in the 1/2MIC to MBC groups (*p* < 0.01). Figure 5H reflects the effect of juglone on the typical Na^+^K^+^-ATPase in *P. aeruginosa*. As shown in Figure 5H, the activity of ATPase in the administration group was significantly lower than that in the control group (*p* < 0.01), indicating that the activity of ATPase seriously restricted the normal growth of *P. aeruginosa*.

### 3.6. Intracellular Oxidative Stress Induced by Juglone

The fluorescence intensity of DCFH-DA was detected by flow cytometry to further evaluate intracellular ROS level changes. As shown in Figure 6, ROS content in *P. aeruginosa* was dose-dependent on the concentration of juglone. Compared with the control group, the reactive oxygen species (ROS) level of the cells treated with juglone (1/2MIC, MIC, MBC) was significantly increased. The ROS content in the MIC and MBC groups was 100 times higher than that in the control group (*p* < 0.01). Reactive oxygen species were also observed in untreated cells, but to a lesser degree. Therefore, when *P. aeruginosa* was exposed to juglone, the intracellular ROS content was significantly increased, resulting in an abnormal intracellular oxidative stress level.

### 3.7. Effect of Juglone on Biofilm Formation of P. aeruginosa

Figure 7A depicts the inhibitory effect of juglone on the formation of *P. aeruginosa* biofilm. The OD value decreased to 0.769, 0.689, 0.344, and 0.166 as the concentration of juglone increased from 1/8MIC to MBC, and the inhibition rates of juglone on biofilm formation were 17.67%, 26.23%, 63.17%, 82.23%, and 92.13%, respectively. Therefore, the biofilm was almost completely cleared at MBC. At the same time, we evaluated the difference in the number of living cells in the biofilm, and we obtained similar results. As can be seen from Figure 7B, the number of living cells was negatively correlated with the concentration of juglone. When *P. aeruginosa* was exposed to 1/8MIC–MIC, the number of living cells decreased by 2.1, 4.3, 6.2, and 7.1 log_10_ CFU/mL, respectively. Figure 7C also shows that juglone significantly reduced the metabolic activity of *P. aeruginosa*, and each treatment group had a significant inhibitory effect compared with the control group.

Considering the significant inhibitory effect of juglone on the formation of biofilm, we further characterized the biofilm by ESEM. As shown in Figure 7D, the cell structure of the control group was normal, with a dense biofilm and typical extracellular polymer structure. In the experimental group, 1/2MIC led to a destruction of the biofilm structure without visible extracellular polymer. Almost no cells with an intact biofilm were observed when *P. aeruginosa* was exposed to juglone at MIC.

We further evaluated the inhibitory effect of juglone on the swimming and swarming movement of *P. aeruginosa* (Figure 7E). The effect of juglone on the colony diameter of *P. aeruginosa* was dose-dependent, and the colony diameter gradually decreased with the increase in juglone concentration. Compared with the control group, the swimming movement of bacteria in the MIC group was significantly inhibited. When the concentration increased to MBC, no obvious colony was formed, indicating that the swimming movement of *P. aeruginosa* was completely inhibited. Similarly, each administration group also showed a significant inhibitory effect on the colony movement of *P. aeruginosa*. With the increase in juglone concentration, the community diameter gradually decreased, and the colony size was gradually reduced.

### 3.8. RT-qPCR Validation

Bacterial extracellular polymers (ESP) are usually listed as important components of biofilms. Bacteria that can synthesize extracellular polysaccharides generally have an ESP gene cluster on the chromosomal genome. The main function of these gene clusters is to encode glycosyltransferase and proteins related to transcriptional regulation, polymerization, and secretion output. EspA and espF are two key genes in the ESP gene cluster. As shown in Figure 8, juglone can effectively block the expression of these two genes, and the inhibition rates of the two genes in the low-concentration MIC group were 36.99% and 47.23%, respectively. Inhibition rates were 47% and 68% in the MBC group.

The strong pathogenicity of *P. aeruginosa* is related to the pathogenic genes it carries. Exou, Exos, Phzm, Toxa, and Lasb control and encode different proteins and enzymes, and the toxins produced can quickly destroy the defense ability and normal function of host cells, eventually leading to organ damage and cell death [19,20]. Therefore, the expression differences of the five genes involved in toxin synthesis at the mRMA level were evaluated by RT-qPCR. As shown in Figure 8, both MIC and MBC groups had a significant hindrance effect on mRNA generation. After MBC treatment, the expression of Exou and Exos genes was most severely inhibited, and their expression rates decreased by 68% and 56%, respectively, compared with the control group (*p* < 0.01). The other three genes, Lasb, Toxa, and Phzm, were also reduced by 44%, 37%, and 47%, respectively. These results showed that juglone may cause cell death by disrupting the gene expression pathway and, thus, blocking the synthesis of the related substances.

## 4. Discussion

Due to the high drug resistance and the common contamination in many foods, it is urgent to develop effective antimicrobial agents against *P. aeruginosa* [19]. Due to its high antimicrobial and anticancer activities, juglone has been widely used as an anticancer drug in clinical practice [11]. Meanwhile, juglone has an obvious antibacterial effect on many foodborne pathogens, spoilage bacteria, and plant pathogens, such as *Escherichia coli*, *Staphylococcus aureus*, *Listeria monocytogenes*, and *Pseudomonas syringae* [13,21]. This study showed that juglone has a dose-dependent inhibitory effect on *P. aeruginosa* (Figure 2).

It is well known that the cell membrane is the basic structure for bacteria to maintain their normal function and resist external invasion [22]. More interestingly, some antibacterial drugs also take the cell membrane as an important target, and they achieve the purpose of sterilization by destroying the integrity and permeability of the cell membrane [23,24]. FE-SEM images indicated that *P. aeruginosa* exhibited significant morphological changes when exposed to juglone. In particular, the biofilm of *P. aeruginosa* was completely destroyed, and the bacteria no longer had a normal physiological structure (Figure 4). SYTO9 and PI are common nucleotide-binding probes used for assessing antimicrobial properties, and they have become powerful tools for assessing cell survival and death [25]. Results obtained by laser confocal microscopy fully confirmed the above statement. As can be seen from Figure 3, juglone can effectively led to impaired cell function and structural abnormalities.

The fluidity of the cell membrane is an important indicator to monitor the normal structure of the cell membrane. Some bacteriostatic agents, such as phenols and polypeptides, have a relatively small molecular weight, which can penetrate cell membranes and cause exudation of the contents [26,27]. DPH can be inserted into a lipophilic phospholipid bilayer without affecting its structure; thus, the DPH fluorescent probe technique was used in this study [18]. When the concentration of juglone reached MIC and MBC, the fluorescence intensity of DPH decreased (Figure 5A), indicating that the number of phospholipids available for DPH to attach to decreased significantly, and the cell membrane fluidity was damaged. This led to a gradual increase in the thickness of the phospholipid membrane, which rendered it more unstable and eventually led to cell atrophy [26]. The mechanism of bacterial membrane damage by this walnut ketone is consistent with Portetel et al. [28], which may be due to the formation of pores, vesicles, and tubules on the surface of the cell membrane by juglone, which ultimately reduces the fluidity of the cell membrane.

Proteins are the basic building blocks of all living things [29]. Figure 5B,C show that the extracellular protein level increased significantly with the concentration of juglone, indicating that it reduces the intracellular protein bands of *P. aeruginosa*. One possible explanation is that juglone may destroy the integrity and permeability of the cell membrane, leading to the leakage of a large number of contents of the cell. Another possible explanation is that juglone blocks protein synthesis in *P**. aeruginosa* and even inhibits the synthesis of related enzymes and DNA expression [16]. Subsequently, the changes in intracellular nucleic acid content were further evaluated. Results showed that the content of intracellular nucleic acids decreased gradually and the content of extracellular nucleic acids increased significantly (Figure 5D–F). The release of nucleic acid resulted in the loss of substantial genetic information and could seriously damage the cell′s ability to self-regulate, via processes such as transcription, translation, and DNA replication [30].

ATP plays an important role in life activities, such as providing energy for most chemical reactions, as well as participation in active transport and biosynthesis [15]. With the destruction of the cell membrane, the ATP in the cell was also leaked (Figure 5G). Another possible explanation is that juglone blocked ATP production and accelerated the activity of ATPase to consume ATP [31]. In addition, juglone inactivated adenosine triphosphate Na^+^ K^+^-ATPase, which was an important factor leading to cell death. In summary, juglone achieved its bactericidal purpose by causing irreversible damage to the cell membrane and cell morphology of *P. aeruginosa*.

As a byproduct of normal biological respiratory metabolism, the body itself maintains low levels of ROS. This allows cells to induce normal oxidative stress in the cell during an emergency, which then stimulates the expression of specific genes that produce more repair proteins in response to cell damage [32]. As shown in Figure 6, the low level of 1/4MIC caused the normal oxidative stress state of cells, whereas the MIC and MBC groups induced a large amount of intracellular ROS production. The accumulation of large amounts of ROS can cause the degradation of intracellular proteins, nucleic acids, and other biological macromolecules, leading to accelerated cell senescence and even death.

Bacterial cell membranes have been reported to enhance their ability to survive in extreme environments, and 80% of human bacterial infections have been linked to biofilms [32]. Therefore, through a quantitative analysis of crystal violet and determination of the number of living cells in the biofilm community, we further verified that juglone not only has a negative effect on the planktonic bacteria, but also has a dose-dependent effect on the inhibition of biofilm formation. At the same time, ESEM observation also confirmed that juglone destroyed the integrity of the biofilm and significantly reduced the number of living cells in the cell membrane.

Swarming is a kind of multicellular motile swarming bacterial behavior, which is related to the formation of biofilm [33]. Results in this study show that juglone effectively reduced the metabolic activity of the bacterial community in the biofilm, as well as restricted the swimming and swarming movement of bacteria (Figure 7E). Hence, juglone prevents biofilm formation by limiting the movement of bacteria.

Extracellular polymers are not necessary for most cells, but they can give some new functions to cells. For example, they can improve the body’s stress response to an extreme external environment, can quickly develop resistance to some alien invaders, can adsorb foreign organic or inorganic substances, and can help cells in the body to adsorb and recognize other cells [34]. In this study, we investigated whether juglone can effectively inhibit the expression of espA and espF genes in the ESP gene cluster. The main function of espA is to determine the length of a polysaccharide chain and to polymerize and secrete repeat units into extracellular cells. Espf is responsible for linking glycosidic bonds between monosaccharides in polysaccharide repeating units. Juglone could effectively inhibit the expression of these two genes, thereby preventing the formation of extracellular polymers and reducing the viability of *P. aeruginosa.* When *P. aeruginosa* infects host cells, it produces a large amount of toxin. The proteins encoded by Exou and Exos play a significant role in pathogenesis; thus, these two genes are considered to be the most important determinants of virulence in *P. aeruginosa* [19]. Exou encodes a cytotoxin that produces phospholipase A2 activity, while Exos encodes an enzyme that inhibits glycosyl transfer [35]. The PhZM gene controls the production of pyocyanin (a blue redox-active phenazine pigment), which can disrupt intracellular calcium homeostasis, inhibit epidermal cell growth, induce neutrophil apoptosis, inactivate α1 protease, etc. [36]. The toxA gene regulates the synthesis of exotoxin A, which interacts with lengthening factor 2 to inhibit the synthesis of related proteins in cells [22,37]. The LASB gene controls the synthesis of elastase, which inactivates immune factors, as well as inactivates human cytokine γ and large amounts of necrosis factor α, in coordination with related proteases [6]. Figure 8 shows that juglone could significantly inhibit the expression of these pathogenic genes, as well as can greatly reduce the infectivity of *P. aeruginosa*.

## 5. Conclusions

This study proved that juglone has a strong inhibitory effect on *P. aeruginosa*. The integrity and permeability of the cell membrane were impaired by juglone, leading to large leakage of the cytoplasm. In addition, juglone could cause abnormal intracellular oxidative stress and accelerate cell senescence and death. At the same time, we further found that juglone could effectively inhibit the expression of toxigenic genes and control the expression of extracellular polymeric genes, thus reducing the infectivity and toxicity of *P. aeruginosa* and blocking the generation of EPS. As a new, natural, and effective bacteriostatic agent of plant origin, juglone has great potential in food preservation.

## Figures and Tables

**Figure 1 molecules-26-05854-f001:**
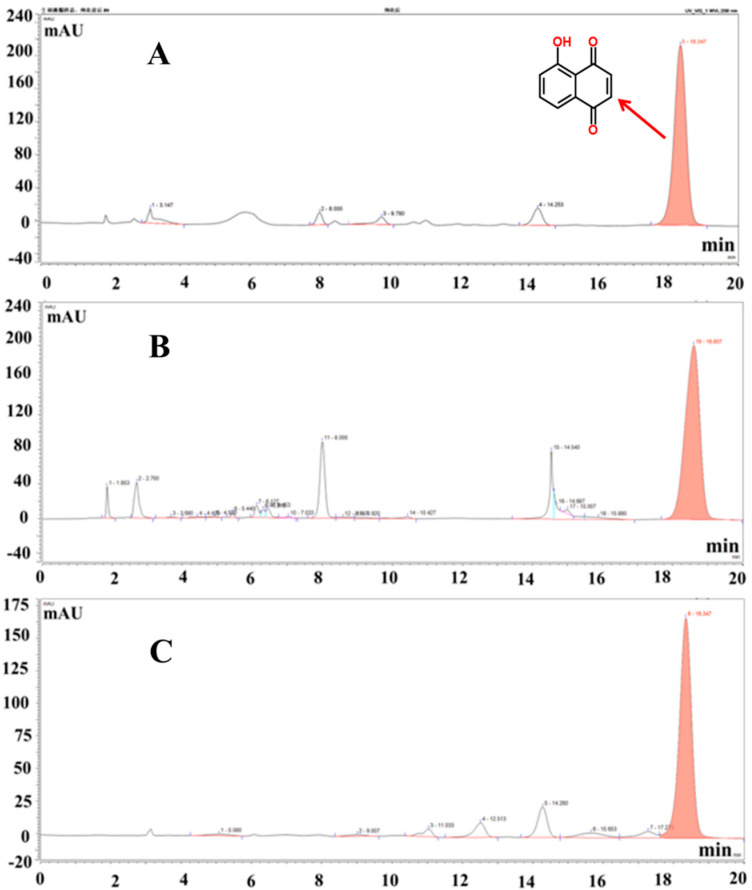
The HPLC chromatograms of the juglone standard (**A**), extracted juglone crude (**B**), and purified juglone (**C**). Elution time is shown on the horizontal axis, and electrical signal strength is shown on the vertical axis.

**Figure 2 molecules-26-05854-f002:**
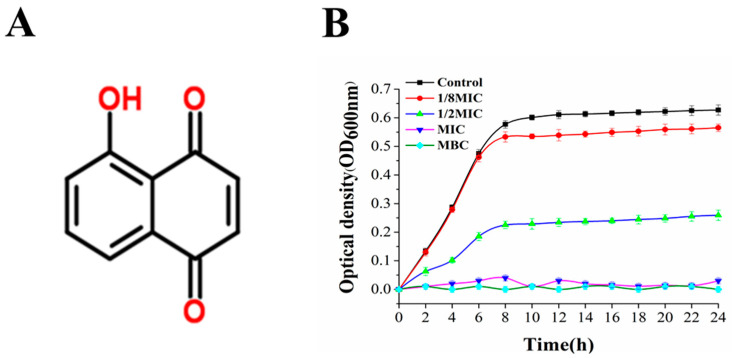
(**A**) Chemical structure formula of juglone; (**B**) growth curve of *P. aeruginosa* exposed to juglone.

**Figure 3 molecules-26-05854-f003:**
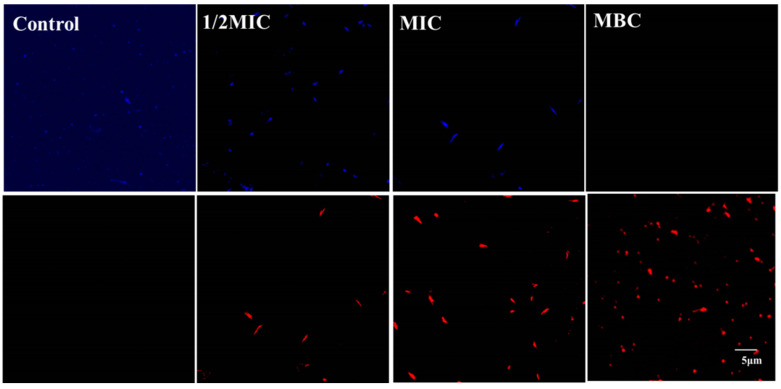
Effect of juglone on cell viability. CLSM (confocal laser scanning microscopy) was used to analyze the viability of *P. aeruginosa* exposed to different concentrations of juglone.

**Figure 4 molecules-26-05854-f004:**
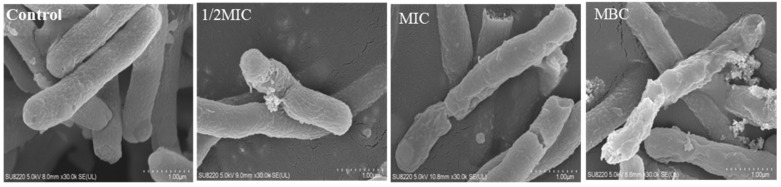
FE-SEM images of *P. aeruginosa* following juglone treatment.

**Figure 5 molecules-26-05854-f005:**
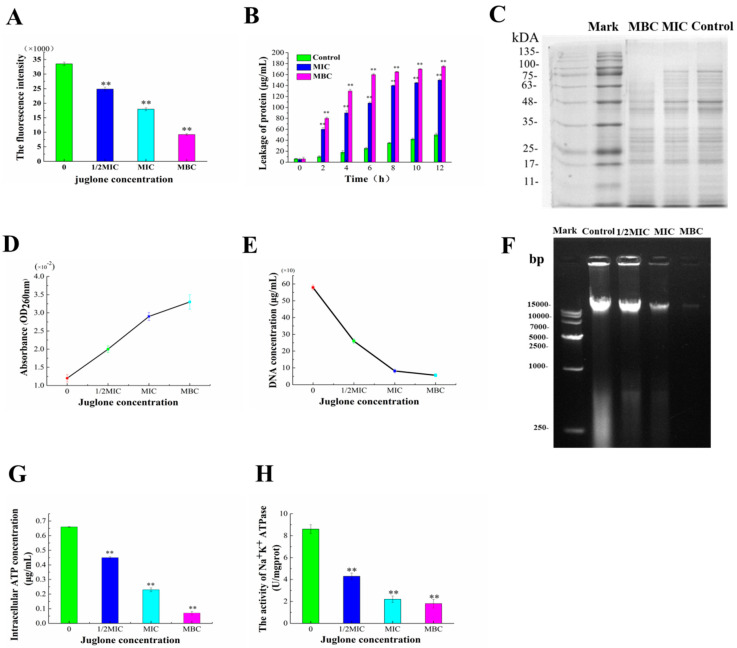
Effect of juglone on the intracellular substances of *P. aeruginosa*. The fluorescence intensity of DPH reflects the degree of cell membrane destruction in *P. aeruginosa* exposed to juglone (**A**). Leakage of extracellular proteins from *P. aeruginosa* treated with juglone (**B**). SDS-PAGE profile of intracellular proteins of *P. aeruginosa* (**C**). Intracellular and extracellular nucleic acid leakage of *P. aeruginosa* treated with juglone (**D**,**E**). DNA gel electrophoresis analysis (**F**). Effect of juglone on intracellular ATP concentration and ATPase activity of *P. aeruginosa* (**G**,**H**). Each bar represents the mean ± SD of three independent experiments; ** *p* < 0.01 versus the control group.

**Figure 6 molecules-26-05854-f006:**
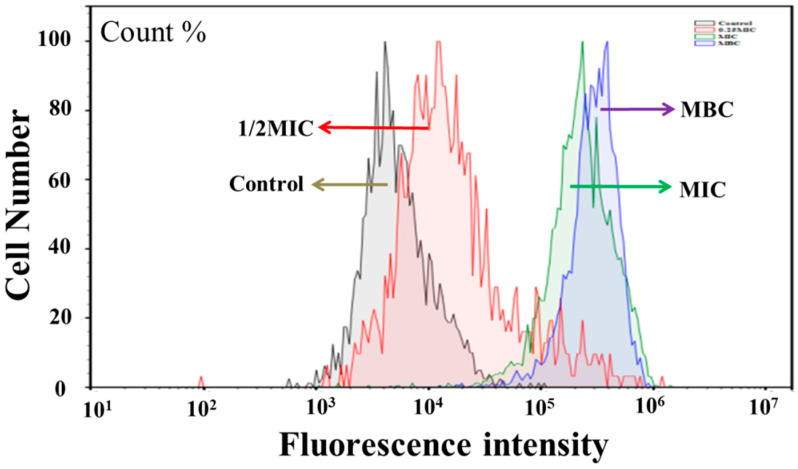
Effect of juglone on the ROS content of *P. aeruginosa*.

**Figure 7 molecules-26-05854-f007:**
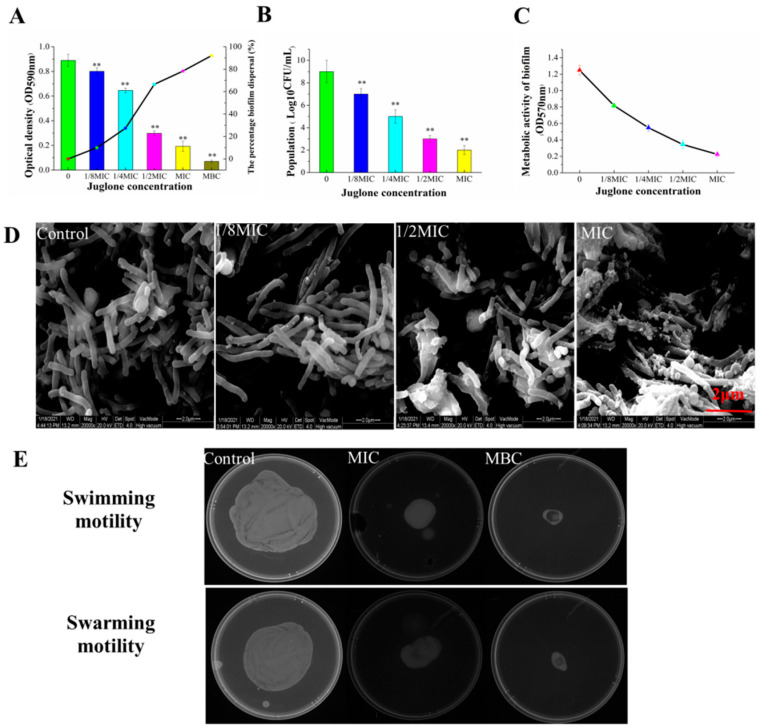
Inhibitory effect of juglone on the biofilm formation of *P. aeruginosa*. Crystal violet quantitative analysis (**A**), viability determination (**B**), cell viability determination (**C**), ESEM image analysis (**D**) and swimming and swarming motility effect of *P. aeruginosa* treated with juglone (**E**). Each bar represents the mean ± SD of three independent experiments; ** *p* < 0.01 versus the control group.

**Figure 8 molecules-26-05854-f008:**
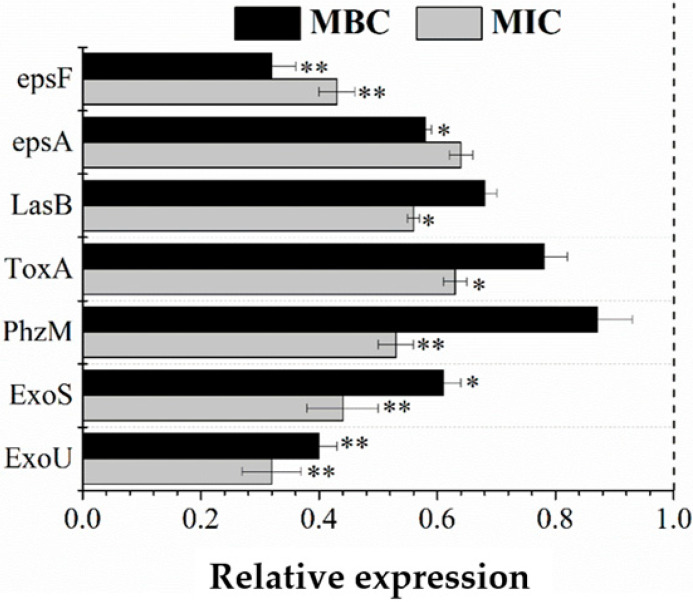
Inhibitory rates of juglone against seven genes. * *p* < 0.05, ** *p* < 0.01.

**Table 1 molecules-26-05854-t001:** Primer sequence.

Gene	Forward Primer Sequence	Reverse Primer Sequence
Exou	CCAACACATTAGCAGCGAGA	TGGGAGTACATTGAGCAGCA
Exos	CATCCTCAGGCGTACATCCT	ATCGATGTCAGCGGGATATC
Phzm	CGGCGAAGACTTCTACAGCT	AGGTAGATATCGCCGTTGGA
Toxa	ATGGTGTAGATCGGCGACAT	AAGCCTTCGACCTCTGGAAC
Lasb	ACATCGCCCAACTGGTCTAC	ACCAGCGGATAGAACATGGT
Ecfx	ATGCCTATCAGGCGTTCCAT	GGCGATCTGGAAAAGAAATG
epsA	TTTATCGATGATACGGTTGCAAG	CTAATAGCCAAGCGGCTCACTC
epsF	CGGCTTTGAACGGTGGG	TCACTGTCCTTCTGCCGCG

**Table 2 molecules-26-05854-t002:** Diameter of inhibition zone, minimum inhibitory concentration (MIC), and minimum bactericidal concentration (MBC).

DIZ (mm)	MIC (μg/mL)		MBC (μg/mL)	
juglone	juglone	Gallic acid	juglone	Gallic acid
22.89 ± 1.00	35	600	45	5000

## Data Availability

Data is contained within the article.

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
