# Peer review of "Juglone Inactivates Pseudomonas aeruginosa through Cell Membrane Damage, Biofilm Blockage, and Inhibition of Gene Expression"

_molecules, 2021, doi:10.3390/molecules26195854_

Round 1
Reviewer 1 Report
The manuscript submitted for review raises a very interesting and important problem. The search for natural antimicrobials and the evaluation of their effectiveness are always topical and necessary issues. Moreover, the authors approached the subject in a very comprehensive manner. Nevertheless, there are a few minor drawbacks in the manuscript that need improvement:
- I suggest changing the title of the work to better reflect the wide range of research done by the authors. Currently, it only suggests a simple assessment of bacterial survival
- I propose to enrich the Introduction with some numerical data on the incidence of P. aeruginosa infections related to food
- Why did the authors conduct research based only on one strain? Why were no food / water isolates included in the study?
- Line 70: Why was this incubation time chosen?
- Line 90: How and in what medium was the bacterial suspension prepared? How was the number of bacteria in the suspension determined?
- Line 105: It should be: Kang et al.
- Line 108: How long was the measurment conducted?
- Line 109: Please state the model and manufacturer of used device
- Point 2.7.1.: n my opinion, it is worth determining the minimum concentration eradicating biofilm and not only the influence of MIC on the biofilm
Author Response
First of all, thank you very much for your constructive suggestions. I believe my article will be improved to a higher level after being revised according to your suggestions.
- I plan to change my title to "Juglone inactivated Pseudomonas aeruginosa through cell membrane damage, biofilm block and inhibit gene expression".
- According to the teacher's suggestion, I have further elaborated the harmful effect of P. aeruginosa in food in the introduction part.
- My explanation for the teacher's first question includes the following two points. First, the main direction of this research group is to find natural bacteriostatic agents. I have done antibacterial tests on common pathogenic bacteria such as Staphylococcus aureus, Escherichia coli, Salmonella and Listeria before, but these studies have been published by many scholars before me, so I did not include them in this article. Secondly, P. aeruginosa has a more complex way to infect food and harm to people than the above bacteria. For example, P. aeruginosa has strong toxicity and drug resistance, and its cell membrane can rapidly make corresponding changes in extreme environment to protect its own safety. Therefore, in order to further explore the antibacterial mechanism of juglone against P. aeruginosa, we chose to study this strain.
- aeruginosa ATCC 10145 strain is a preserved strain isolated from food by our team, which can be directly used in experiments.
- According to liu et al. study, P. aeruginosa generally reached its maximum growth concentration after 16 h of growth.
- Prior the experiment, the activated P. aeruginosa strain was cultured in MH broth at 37℃ for 16 h in shaking flask, and diluted with MB broth until the OD value was approximately 0.5 (approximately 107 CFU/mL). Corresponding modifications have been made in line-90.
- Corresponding modifications have been made in the original place.
- The test tube was taken out every 2 h for 24 h, and the OD (wavelength of 600 nm) value of bacteria was measured by spectrophotometer (1530, Thermo Fisher Scientific Oy).
- The model and manufacturer of the device used have been added.
- The removal of biofilm at MBC concentration has been further studied. The results showed that the biofilm removal rate reached 92.13%, which can be said that the biofilm was completely removed when the concentration of juglone was MBC.

Reviewer 2 Report
The work is well done, however, despite the fact that you conducted numerous studies, the discussion is a bit superficial, I think you could try to dig a little deeper, perhaps better explaining what each study was done for and as a whole what implications it has.

Author Response
I have revised all the questions in the article according to the teacher's requirements and suggestions. Thank you very much for your constructive suggestions. I believe my article will be improved to a higher level after modification according to your suggestions.
Reviewer 3 Report
The manuscript by Youlin Zhang and Qiqi Han reports the plant-derived Juglone inhibitory effect on the growth of Pseudomonas aeruginosa. This study elucidated the inhibitory effect of juglone on the growth of plankton and P. aeruginosa biofilm eradication. The authors highlight P. aeruginosa colonies inhibition when exposed to juglone (35 μg/mL). Also, blocks the expression of 5 virulence genes, 2 genes involved in the production of extracellular polymer, leading to a reduction in toxicity and infection of P. aeruginosa and preventing the production of extracellular polymers. However, in this reviewer's opinion, the authors need to justify the following:
The authors proceeded with biological experiments with 70% pure Juglone. They could have done these experiments with inexpensive commercially available Juglone with the highest purity to validate these results.
The manuscript contains significant grammatical, spacing and typo errors:
Few examples are:
-Introduction: line 26
-Abstract: line 15
Line 136
Line 144
Line 160
Line 170
Line 421
Line 432
Line 443
Line 458
Line 463
Line 499
Line 504
etc.
Author Response
First of all, thank you very much for your constructive suggestions. I believe my article will be improved to a higher level after being revised according to your suggestions.
It is well known that juglone, as a plant - derived bacteriostatic agent, has broad - spectrum bacteriostatic activity. In China's Shaanxi province, 3.2 million tons of walnuts are produced each year, resulting in about 400,000 tons of walnut husk, which will cause some damage to the environment. But walnut husk contains a lot of juglone, so extracting juglone from walnut husk waste can not only solve the environmental impact of walnut husk, but also produce certain economic benefits for farmers. This is the dual purpose of this study, to explore the extraction and purification process of juglone, and also to find an effective antibacterial agent against P. aeruginosa.
As for the grammar of the sentence pointed out by the teacher, I have modified it one by one in the article. Thank you again for your valuable advice.